An improved method for preparing stained ground teeth sections

Qin Qizhong 1
Li Yueheng 2
Zhou Yujian zyujian@cqmu.edu.cn 1
1 Experimental Teaching and Management Center, Chongqing Medical University , Chongqing , Yuzhong district , China
2 Stomatological Hospital of Chongqing Medical University; College of Stomatology, Chongqing Medical University; Chongqing Key Laboratory of Oral Diseases and Biomedical Sciences; Chongqing Municipal Key Laboratory of Oral Biomedical Engineering of Higher Education, Chongqing Medical University , Chongqing , Yubei District , China
Tomar Mahendra
Electronic publication date: 2023 Apr 28
Publication date: 2023
Volume: 11
Electronic Location ID: e15240
Received 2022 Sep 20; Accepted 2023 Mar 28
Copyright: ©2023 Qin et al.
Copyright year: 2023
Copyright holder: Qin et al.
License: This is an open access article distributed under the terms of the Creative Commons Attribution License, which permits unrestricted use, distribution, reproduction and adaptation in any medium and for any purpose provided that it is properly attributed. For attribution, the original author(s), title, publication source (PeerJ) and either DOI or URL of the article must be cited.
License URL: https://creativecommons.org/licenses/by/4.0/

Keywords: Ground sections of teeth, Rosin, Un-demineralization

Funding: Education Commission of Chongqing Municipality KJ1500211 First Batch of Key Disciplines On Public Health in Chongqing; College of Stomatology, Chongqing Medical University KQJ202205 This study was supported by the Education Commission of Chongqing Municipality (KJ1500211) and the First Batch of Key Disciplines On Public Health in Chongqing; College of Stomatology, Chongqing Medical University (KQJ202205). The funders had no role in study design, data collection and analysis, decision to publish, or preparation of the manuscript.

==============================
Objective

In oral histopathology teaching and research, there is a need for high-quality undemineralized tooth sections that are easy to handle, have controlled thickness, allow the observation of intact microstructures, and can be preserved for long periods of time.

Methods

Teeth were collected under non-demineralizing conditions. Tooth sections (15–25 µm) were prepared using a diamond knife, then randomly divided into three groups: (1) stained with rosin, (2) stained with hematoxylin and eosin, or (3) not stained. The prepared tooth sections were evaluated by microscopy for clarity and microstructure visibility.

Results

The use of a diamond knife in the sectioning and grinding process yielded high-quality ground sections of teeth. Rosin-stained ground sections allowed better identification of microstructures within the teeth, compared with unstained or hematoxylin and eosin-stained ground sections.

Conclusion

The best results were obtained in the ground sections of teeth that were stained with rosin. Ground sections of teeth prepared using this staining method could be useful in oral histopathology teaching and research.

Introduction

In oral histopathology teaching and research, high-quality tooth sections that clearly display microstructures within the teeth are essential for disease diagnosis, monitoring, and comparative analysis (Prasad & Donoghue, 2013). Methods of preparing these tooth sections include decalcification, embedding, sectioning, manual grinding, and diamond knife grinding; each method has unique advantages and disadvantages, and their scopes of application considerably differ. Tooth enamel is primarily composed of inorganic substances, consisting of 93.6%–98.5% mineral content and 0.05%–0.8% organic content (Little, Cueto & Rowley, 1962). Partial decalcification (Liem & Jansen, 1982) is an important method for observing the histological structures of teeth and for examining tooth enamel and other hard tissues. This process involves the preparation of thin sections through grinding or the use of a hard tissue microtome (Donath, 1995; Suvarna, Layton & Bancroft, 2013). Vikhe et al. (2020) reported that tooth hard tissue could be clearly examined after using borite as a grinding aid followed by the Pravara staining process. These conventional methods of preparing ground sections are time-consuming, laborious, involve complicated procedures, and result in sections with nonuniform thickness.

Thus, there is a need for high-quality, undemineralized tooth sections that are easy to handle and have uniform thickness to allow clear observation of intact microstructures, and that can be preserved for long periods of time. The present study was conducted to identify a simple method for preparing ground sections of undemineralized teeth that can facilitate the observation efforts necessary for oral histopathology teaching and research.

Material and Methods

Sample selection

Premolars and impacted teeth that were extracted for orthodontic purposes from patients attending Oral and Maxillofacial Surgery Clinic, the Stomatological Hospital of Chongqing Medical University were used in this study. After tooth extraction, the soft tissues were cleaned and the teeth were inspected. Teeth that were intact (i.e., without visible cracks, white spots, or caries) were stored in 10% formalin for 24 h before use. The Stomatological Hospital of Chongqing Medical University granted ethical approval to carry out the study within its facilities and written consent was obtained from the participants before tooth extraction. The ethical approval was granted on May 15, 2018, with approval number CQHS-REC-2018 (LS No. 22).

Teeth were collected under non-demineralizing conditions. Tooth sections (15–25 µm) were prepared using a diamond knife, then randomly divided into three groups: (1) stained with rosin, (2) stained with hematoxylin and eosin, or (3) not stained. The prepared tooth sections were evaluated by microscopy for clarity and microstructure visibility.

Instruments and reagents

This study used the following instruments: a XQPI-66A ¢ diamond microtome (Changzhou Dedu Precision Instrument Co., Ltd., Jiangsu, China), a CX23LEDRFS1C microscope (Olympus, Tokyo, Japan), a JC-9030A/JC-9030AE oven (Qingdao Jingcheng Instrument Co., Ltd., Qingdao, China), a NAKANISHI high-precision grinding machine (Shenzhen Chunheng Tools and Instrument Co., Ltd., Shenzhen, China), and a P28NM55 cubic boron nitride honing stone (Zhengzhou Boson Abrasives Co., Ltd., Zhengzhou, China). The reagents used in this study were: formaldehyde, anhydrous ethanol, sodium carbonate, xylene, glycerol (Chongqing Chuandong Chemical (Group) Co., Ltd., Chongqing, China), and neutral gum (Shanghai Qiangshun Chemical Reagent Co., Ltd., Shanghai, China). Shellac was used as an adhesive.

Sectioning and staining

Tooth specimens were cut into 200–300-µm-thick sections using a diamond knife operating at 6,000 rpm on a XQPI-66A ¢ diamond microtome. Tooth sections were then ground on a grinding machine: one side of each section was firmly attached to the stainless-steel carrier plate using baking shellac, and the carrier plate had a parallelism tolerance of 10 µm to ensure uniform section thickness. The side of each tooth section that was not attached to the carrier plate was finely ground with a fine honing stone on the grinding machine. Each section was then removed from the carrier plate, placed in 5% sodium carbonate at 60 °C for approximately 30 min to remove the shellac, and then rinsed with water. The other side of the tooth section was then attached to the carrier plate and the same process was followed to grind the other side of the tooth section. After both sides were ground, the final thickness of each tooth section was 15–25 µm.

Each ground section was removed from the carrier plate, cleaned, and dehydrated by incubation with absolute ethanol (twice for 30 min each).

Next, the sections in the rosin-stained group were then stained, as follows: a rosin mixture (100 g of rosin, 20 ml of xylene, and 5 ml of glycerol) was heated in a 100 °C oven until completely melted; tooth sections were then fully soaked in the melted rosin, which resulted in rapid staining; the stained sections were immediately placed on glass slides, mounted with neutral gum, and observed by microscopy.

After the grinding was completed, the tooth sections in the hematoxylin and eosin-stained group were rinsed for five minutes, then immersed with 70% alcohol for 1 h to thoroughly remove impurities. Each tooth section was then mordanted for 10 s and stained with hematoxylin for 5 min, rinsed for 3 min, and then stained with eosin for 1 min. The tooth sections were then dehydrated (with alcohol 95% 1 s, alcohol 100% 4 ×1 s), made transparent (xylene 5 × 20 min), and sealed with neutral resins.

Results

Table 1 shows the tooth structures that were visible in the longitudinal and transverse ground sections after preparation as described in the methods section. Some special structures were clearly visible after rosin staining (Fig. S1). The enamel and dentinal microstructures were compared in the ground sections of teeth according to staining method, as shown in Figs. 1 and 2.

Table 1 Structures visible in ground, rosin-stained sections of teeth.

Type of section	Observation sites	Structures visible in ground sections via microscopy	
Longitudinal ground sections	Dental enamel	Enamel rods, enamel spindles, enamel tufts, Enamel-dentinal junction, incremental lines, enamelo-cemental junction, Hunter–Schreger bands.	
Dentin	Dentinal tubules, interglobular dentin, Tomes’ granular layer	
Cementum	Dentino-cemental junction, cementum lacunae, lamellar phenomena	
Transverse ground sections of the crown	Dental enamel	Incremental lines, enamel lamella	
Dentin	Dentinal tubules, interglobular dentin, Tomes’ granular layer	
Transverse ground sections of roots	Dentin	Dentinal tubules, interglobular dentin, Tomes’ granular layer	
Cementum	Lacunae, dentin tubules, lamina, cementum	

Figure 1 Ground sections of undemineralized teeth by staining method.

*, Dentin tubules; →, Enamel-dentinal junction; ˆ, Thomas particles; &, enamel. Scale bar = 100 µm.

Figure 2 Ground sections of undemineralized teeth by staining method.

*, Dentin tubules; →, Enamel-dentinal junction; #, enamelo-cemental junction; &, enamel; ↓ Dentino-cemental junction. Scale bar = 100 µm.

The dentin tubules, enamel, and other special structures of the tooth were more easily observed in the rosin-stained sections than in the sections stained with hematoxylin and eosin and the unstained sections. For example, the dentin-enamel junction was more easily identified in the rosin-stained tooth sections than in the sections stained with hematoxylin and eosin (Figs. 1 and 2).

Discussion

Oral histopathology is a morphological discipline that involves analyses of the histological structures in dental tissues. Practical instruction is central to oral histopathology education. Some research institutions and universities are still using ground sections of teeth that were purchased in the 1970s and 1980s. Teeth sections are susceptible to gradual wear and tear, so these older teeth sections can hinder effective practical instruction. Because it is difficult to purchase ground sections of teeth, replenishing these resources is a significant problem in oral histopathology education.

Undemineralized tissue sectioning is a common method for studying bone tissues and bone grafting materials (Wang & Liu, 2009), but conventional undemineralized tissue sections cannot easily distinguish the interface between bone grafting materials and implants. Conversely, manually ground sections allow the observation of newly formed bone tissues, and provide insights that cannot be obtained from conventional tissue sections (Geng, Su & Xu, 2005). The current methods of grinding teeth sections include resin embedding, manual grinding, and diamond knife grinding (Cheng et al., 2019; Ho et al., 2004; Zanchi et al., 2010).

As noted in the introduction, each of these methods has unique advantages and disadvantages, and their scopes of applications considerably differ. For example, the resin embedding method uses a resin solution, rather than paraffin. Solidified resin specimens are resilient and cannot be broken easily, making this method appropriate for sectioning and analyzing undemineralized bone tissue sections, but the process, which includes fixation, dehydration, infiltration, and grinding, is both complicated and time-consuming. Sections prepared with a saw are thick, allowing only limited observation of microstructures because of cell overlap (Arbabzadeh et al., 1998). An analysis of three histological tooth preparation techniques found that when using decalcified sections to study hard tissue (e.g., teeth), paraffin embedding and hematoxylin-eosin staining are consistently easier, more economical, and more reliable, compared with techniques that involve resin embedding (Keklikoglu & Akinci, 2013). Another study used precision microsectioning to make 100 ± 20 mm grinding sheets (Babar & Gonzalez, 2011; Silva, Moreira & Alves, 2011). Although this method enables the observation of some structures, it lacks accuracy and hinders microstructure visibility. Manual grinding is time-consuming and results in prepared sections with nonuniform thicknesses. Silva, Moreira & Alves (2011) present a detailed cutting/grinding protocol for isolated human teeth that can be easily reproduced in any research or teaching support laboratory. As noted previously, there is a need for high-quality, undemineralized tooth sections that: are easy to handle, have uniform thickness, allow clear observation of intact microstructures, and can be preserved for long periods of time.

Proper specimen alignment prior to sectioning is important for ensuring that the sections contain the entire region of interest and for ensuring standardized histomorphometric evaluation. Sectioning should be done perpendicular to the long axis of a target component/structure (e.g., a metal implant or a tooth root), or the target object can be distorted in the resulting section (Vikhe et al., 2020; Parfitt et al., 1987).

Longitudinal ground tooth sections have numerous applications in oral histopathology because they allow full observation of most dental structures (e.g., enamel, dentin, and cementum; Jiang, 1956). Hunter–Schreger bands in enamel are unique phenomena that can be observed in longitudinal ground sections of teeth (Jiang, 1956). Longitudinal ground sections of maxillary and mandibular teeth (5-5) can be made in the labiolingual plane by grinding from mesial to distal surfaces. Because of anatomical limitations, longitudinal ground sections of mandibular teeth (6-8) can only be made in the mesiodistal plane by grinding from buccal to lingual surfaces (Jiang, 1956). Maxillary teeth (6-8) usually have three roots, which cannot all be ground on a single plane, so longitudinal ground sections are obtained across the center of each tooth either along the mesiobuccal and palatal root plane or the distobuccal and palatal root plane (Jiang, 1956).

Transverse ground sections of the crown are used to observe structures such as enamel tufts, enamel lamellae, and enamel hyperplasia. Transverse ground sections of roots are used to observe structures such as cementum lacunae, dentinal tubules, and lamellar phenomena (Jiang, 1956).

In the present study, we found that tooth structures were more easily identified in images of rosin-stained undemineralized teeth (Fig. 1), compared with hematoxylin and eosin-stained teeth (Fig. 2). This is likely because structures with limited mineralization and high organic matter content have a greater affinity for pigment compared with highly mineralized areas, making these less-mineralized structures more visible and easily identifiable when stained with pigment-containing rosin. Vikhe et al. (2020) used the Pravara (a plant stain) staining process for analyzing the ground sections of teeth, with the expectation that hard tissue visibility would be improved with this process. However, the authors concluded that the grade of stained sections and the maintenance time of the staining effect should be compared in subsequent studies as their study did not compare these variables when considering this process (Vikhe et al., 2020).

The present study was conducted to address technical problems in the current grinding and staining procedures of ground tooth sections. Although rosin contains natural pigments, that does not guarantee that it can be successfully used to stain bone sections with sufficient quality for use in teaching and research. Thus far, conventional dyes (e.g., hematoxylin-eosin, basic or acid fuchsin, thionine, toluidine blue, fast green, and silver staining reagents) have been used to stain bone sections, but these dyes tend to obscure the internal structures of bone sections and are easily decolorized (Goodwin & Jerome, 1987; Li et al., 2000). Specifically, these dyes tend to obscure enamel structures in tooth sections, such as cross-striations in enamel rods, enamel spindles, enamel tufts, incremental lines in enamel, and differences in density between interrod enamel and enamel rods. Conventional dyes also tend to obscure dentinal structures in tooth sections, such as interglobular dentin, enamel-dentinal junction, incremental lines in dentin, and differences in density between peritubular and intertubular dentin. Laminar structures in cementum are also not clearly visible in tooth sections that have been stained with conventional dyes (Sakoolnamarka et al., 2002; Pai et al., 2009). One study found that modified Gallegol staining could be used to successfully distinguish various types of oral hard tissue in decalcified sections, soft tissue sections, and sectional sections. This type of staining also increases the understanding of lesion characteristics, so it is sometimes used for diagnosis or diagnostic analysis (Sandhya et al., 2014). Tooth sections stained with rosin enable the observation of the structures of enamel, dentin, and cementum that are hard to see in tooth sections stained with conventional dyes. Additionally, after ground sections have been stained with rosin, tooth structures exhibit a three-dimensional appearance (Aldas et al., 2021); they are also more transparent and less likely to be decolorized (Kugler et al., 2019).

In the present study, we investigated the use of rosin to stain ground sections of teeth. Rosin staining was chosen because the refractive index of rosin is 1.527 (Oivanen et al., 2021), which is slightly lower than the refractive index of enamel (1.63; Hariri et al., 2013), and the Abbe number can be increased to reduce dispersion. Because rosin contains a certain amount of natural pigment, melted rosin has good permeability (Carter et al., 2000; Leite et al., 2020). After ground sections of teeth have been soaked in melted rosin, partial pigmentation is visible in multiple tooth structures. Additionally, rosin staining can increase bone transparency, providing superior structural visibility compared with unstained bone. Our findings show that rosin-stained ground sections of teeth facilitate the observation of microstructures within the teeth, while enhancing the textures of dental enamel, dentin, and cementum. These rosin-stained sections could be useful in oral histopathology teaching and research.

Conclusions

Preparing ground sections of teeth by sectioning them with a diamond knife and then staining them with rosin made prominent substantive features of the teeth more visible, improving upon current methods. This method has application value in histopathology teaching and research.

Supplemental Information

Figure S1 Images of rosin-stained undemineralized tooth ground sections

Scale bar = 100 µm.

Click here for additional data file.

Additional Information and Declarations

Competing Interests

Author Contributions

Data Availability

The authors declare there are no competing interests.

Qizhong Qin conceived and designed the experiments, performed the experiments, analyzed the data, authored or reviewed drafts of the article, and approved the final draft.

Yueheng Li conceived and designed the experiments, performed the experiments, analyzed the data, prepared figures and/or tables, authored or reviewed drafts of the article, and approved the final draft.

Yujian Zhou conceived and designed the experiments, performed the experiments, authored or reviewed drafts of the article, and approved the final draft.

The following information was supplied regarding data availability:

The raw data is available at figshare: Qin, Qizhong (2022): An improved method for preparing stained ground sections of teeth. figshare. Figure. https://doi.org/10.6084/m9.figshare.21521505.v1.

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
