# Peer review of "An improved method for preparing stained ground teeth sections"

_PeerJ, doi:10.7717/peerj.15240_

## Round 0.1 · original submission · Major Revisions

Dear Dr. Zhou,
As per the reports of our expert reviewers, the manuscript covers simplified method and is appreciated for quality. However, the manuscript has many issues which need to be addressed. The language of the manuscript should be improved for which I advise taking help from some language editors. Other technical comments may be considered/justified (in case the authors do not agree with the reviewers).

Please revise and resubmit asap.

Good luck.

Reviewer 1 ·

Basic reporting

1-An english revision is required for the whole manuscript.
2-Discussion section is not provided with necessary citations. The entire section had only 4 citations while numerous information which needs citations were presented.
3-A table was presented in the manuscript. That should be presented as Table 1 and should be revised.
4- Citations are sometimes given as "(1), (2)" and sometimes given with the author's names. Please check the author's guideline of the journal and revise the citation format bot in text and in the references section.
5-Materials and methods section should entirely be revised. Authors should describe the methodology they performed in this section. Yet, in the manuscript we see a section which is more like an introduction or discussion section.
6-It would be much better if authors could place some marks and/or labels to the figures indicating specific structures of interest.
7-Each photograph should have a separate bar and the length of the bar should be stated.
8-It could be beneficial to present pictures from rosin stained and unstained sections together to allow us make comparisons between the methods much easily.

Experimental design

1-Is it possible to apply tissue stains such as H&E or trichrome staining techniques on the sections prepared with this method? If so, it would add value to your manuscript to make a simple H&E staining and represent the results.

Validity of the findings

The procedure should be described better in the materials and methods section. Results are well provided.

Additional comments

.Authors should make revisions regarding the english language and the manuscript strucre as materials and methods section is not appropriate and the discussion section lacks of needed citations.

Annotated reviews are not available for download in order to protect the identity of reviewers who chose to remain anonymous.

·

Basic reporting

The manuscript entitled “An improved staining method for preparing tooth ground sections” was aimed at providing a method of preparing undemineralized tooth ground sections with simplified procedures, and improved results from the tooth ground sections, for teaching and scientific research.

The authors are appreciated for their excellent work in this area; however, the presentation of results needs to be improved to have a clear message. Suggestions have been provided section wise.

Experimental design

Abstract: in the methods section, the sentences are to be reframed to have a clear message. Like teeth were divided into two groups, one group was stained with rosin stain while the other group contained unstained. Sectioning was performed using an ultra-cut microtome with a diamond knife.
Introduction
The authors have highlighted the need for such sectioning and staining methods. Staining with rosin stain is a novelty in this manuscript. However, it is also suggested to include the possible advantages of the present study over the already available staining and sectioning methods of teeth samples (Plant-based staining method by Vikhe et al., 2020; Modified Gallego’s stain by Sandhya et al., 2014; hematoxylin and eosin staining by Keklikoglu and Akinci 2013).
Results
The findings of the experiment should be written in sentences citing the figures in brackets to have a clear understanding of the findings. The results are written as if they are legends of the figures. The legends of figures should be written separately.
Figures
Please add labels of structures visible in the figures and add them to legends too. Scale is missing in the figure except for one of them in both the figures (1&2). It is not clear whether the scale bar is the same for all the photographs in the panel, if so, please write the scale bar is the same for all the figures.

Validity of the findings

Discussion
The discussion is mainly based on the available books and techniques for decalcification and sectioning of bones. There are several research papers related to sectioning and staining of sections available in the public domain to discuss the results (Plant-based staining method by Vikhe et al., 2020; Modified Gallego’s stain by Sandhya et al., 2014; hematoxylin and eosin staining by Keklikoglu and Akinci 2013). The basic principle behind this particular staining is its refractive index and permeability whether the available stains have similar or different parameters.

A few suggested references for Introduction and Discussion are:
Vikhe, D. M., Thete, S. G., Mastud, C. S., Mantri, T., Mhaske, P. N., Mastud, S. P., & Madanshetty, P. (2020). Staining the Ground Section of the Tooth Using an Innovative Plant Stain Found in the Pravara Region, India. The Journal of Contemporary Dental Practice, 21(10), 1114.
Sandhya, T., Avinash, T., Srivastava, C., Satheesan, E., & Bhalerao, S. (2014). Modified Gallego’s stain as a differential stain for oral hard tissues in oral pathology: A preliminary report. Int J Oral Maxillofac Pathol, 5, 2-6.
Silva, G. A. B., Moreira, A., & Alves, J. B. (2011). Histological processing of teeth and periodontal tissues for light microscopy analysis. In Light Microscopy (pp. 19-36). Humana Press, Totowa, NJ.
Keklikoglu N, Akinci S. Comparison of three different techniques for histological tooth preparation. Folia Histochem Cytobiol. 2013;51(4):286-91. doi: 10.5603/FHC.2013.0039. PMID: 24497133.
Babar, M. G., & Gonzalez, M. A. (2011). An Alternative Efficient Technique for Thin Tooth Sectioning. International e-Journal of Science, Medicine and Education, 5(1), 27-30.

Additional comments

Authors are suggested to read the text and correct few grammatical errors (in the form of usage of aarticles) to improve its standard.

·

Basic reporting

The research manuscript is very well written with grammatical mistakes and with good in depth analysis of the content under study. All the chapters of the manuscript are self explanatory and unambiguous. Hence, the current manuscript may be accepted for publication as it is without any modification or correction.

Experimental design

Very correct and properly explains the purpose of the study.

Validity of the findings

The findings are quite useful for teaching ang research purposes.

Additional comments

The current reviewed manuscript can be accepted for publication as it is.

---

## Round 0.2 · Minor Revisions

Dear Dr. Zhou,

As the corrections made by you during previous revisions were checked by our expert reviewers and they really appreciated your efforts to match the journal’s standards. However, one of our reviewers feels that still a few more points need to be addressed to improve the quality of the manuscript.

Therefore, as the Academic Editor for your article I wish you to invite for Minor Revisions.

My suggested changes and reviewer comments are shown below and, on your article, 'Overview' screen.

Please address these changes and resubmit. Although not a hard deadline please try to submit your revision within the next 21 days.

Regards and good luck with your future submissions.

Reviewer 1 ·

Basic reporting

1- Authors should place a space before using a paranthesis.

2- The last paragraph of the "Materials and Methods" section needs a language revision.

3- The former criticism on the lack of citations continues in this revised manuscript. Authors did put an effort to add some other information and placing citations for them. However, there are stil so many knowledge are missing citations. I have indicated most of them which could be seen in the added file. Authors are recommended to add citations for these informations, where applicable.

4- The materials and methods section should only describe what did authors perform. After explaining the sample selection, you can briefly say, longitudinal and transverse ground sections were taken to see specific details in the tissue. Then you can continue with describing how you take these sections.
The entire explanations given for the longitudinam and tranverse ground sectioning are not approppriate for materials and methods section. Authors can choose to briefly describe it's importance in discussion section if they'd like.

5- In the figure 2, D; there's a problem with the arrow at the top left side of the tissue. It's a different arrow than authors used to describe dentinoenamel junction and has a white rectangle, blocking the view, around it.

Experimental design

Research is well defined and the problem is well presented. Methodology is sufficient.

Validity of the findings

1- The results were well presented in figures and in the table. Still, the section could be enriched. By looking the figures, we can say that all of the specific structures were visible in both samples. But it was harder to identify enamel in HE stained sections, dentin tubules had much better visuals in rosin stained sections and dentinoenamel junction was much more identifiable in rosin and HE stained sections compared to non-stained sections. Authors should describe these findings in detail, within this section.These findings also should be discussed in the discussion section by making clear inter-sample comparisons.

2- The discussion section is almost entirely based on the need of a new and better technique by describing the limitations of other techniques, etc. In this section, after a brief explanation of these needs and limitations, authors should discuss their findings. What differences did authors reveal after their study between the used methods? What are the improvements that their technique brings into the table comparing the traditional techniques? If there are other attempts to find a better technique, what comparisons can be made between these attempts and this particular study? I recommend authors to rebuild their discussion, entirely.

Additional comments

Authors had put a great effort on revising the manuscript. Yet, there are still some improvements to make in order to prepare the study for publishing. Some little parts of the manuscript still needs language editing and both materials and methods, and discussion section needs revisions. The reviewed manuscript could be seen in the attached file for details.

Annotated reviews are not available for download in order to protect the identity of reviewers who chose to remain anonymous.

·

Basic reporting

Authors of the manuscript entitled “An improved method for preparing stained tooth ground sections” have taken sincere note of the referee’s comments. They have revised the manuscript accordingly.
Necessary corrections have been made by rewriting the sentences to have a clear message. Grammatical errors have been corrected.

Experimental design

Necessary corrections have been made.

Validity of the findings

Suggested references have been added in the introduction and discussion. This has improved the scientific message.
Labels of structures visible in the figures have been added. Scales have been placed on the figures.
The discussion has been improved by the inclusion of available references.

---

## Round 0.3 · Minor Revisions

Dear Authors,

One of our reviewer could not see the modifications/revisions made by you in the file resubmitted. Please do the needful as asked by reviewer and resubmit it asap.
Best of luck

Reviewer 1 ·

Basic reporting

Please make the revisions by using "track changes" tool of MS word, instead of making sidenotes that you made changes. I cannot see what you changed and how you changed. Please use this tool, delete what you want to delete and write what you want to write and re-send the manuscript. I cannot see any tracked changes other than the placed citations. I should see how it was prior to your revision and how it looks now. There are just side notes indicating that part is revised or taken to the discussion section.

Experimental design

.

Validity of the findings

.

Additional comments

Please prepare an appropriate document which I can see what revisions you had made and re-upload your manuscript for reviewing.

---

## Round 0.4 · Minor Revisions

Dear authors,

As per continuous criticism from our reviewers, the manuscript still does not match our quality. The major objection is about inadequate citations supporting the statements and the language in certain places is not suitable as also mentioned in the comments of the reviewer.

I will suggest you take help from a professional language editor to match our standards.

Please do the needful and resubmit asap.
Best of luck.

Reviewer 1 ·

Basic reporting

1) The used language had been ciriticised in the last 2 rounds yet, still we see an unprofessional use of English. Some of the examples could be seen below:
-"After the grinding procedures were completed, rinse it for 5 minutes, thoroughly remove impurities, immerse it with 70% alcohol for 1 hourmordanting 10 seconds and stain with hematoxylin for 5 minutes ,then rinse 3minutes, and stain with eosin for 1 minute, dehydration (alcohol 95%, alcohol 100%, alcohol 100%, alcohol 100%, alcohol 100%) every 1 second, transparent (dimethylbenzene (1), dimethylbenzene (2), dimethylbenzene (3), dimethylbenzene (4), dimethylbenzene (5)) each 20 minutes, seal with neutral resins."

- "The staining increased the understanding of lesion characteristics, so it was sometimes advantageous to use it for diagnosis or diagnosis analysis"

-"But it was harder to identify enamel in HE stained sections, dentin tubules had much better visuals in rosin stained sections and dentinoenamel junction was much more identifiable in rosin and HE stained
sections compared to non-stained sections".

2) Again, despite the criticisms abour the uncited informations given in the manuscript, authors still adding new information without proper citations in this 3rd review.

Please see the annotated pdf file in the attachment.

Experimental design

No comment.

Validity of the findings

No comment.

Additional comments

I really do want to suggest an acceptance for your manuscript to the editor but before that, authors should improve their technical language. You cannot use uncited information in your text. For a journal of PeerJ's level, you should not use poor terminology either. Please revise your terminology, add proper citations and get profesional editing services if needed.

Regards.

Annotated reviews are not available for download in order to protect the identity of reviewers who chose to remain anonymous.

---

## Round 0.5 · accepted · Accept

Dear Dr. Zhou,

It is my pleasure to inform you that as per the recommendation of our expert reviewers, the manuscript "An improved method for preparing stained ground teeth sections" - has been Accepted for publication in PeerJ.

This is an editorial acceptance and you will be intimated for the list of further tasks before publication. So, I request you to be available for a few days to make the necessary things asap.

Regards and good luck with your future submissions.

Reviewer 1 ·

Basic reporting

Thank you for your improvements. I have no further inquiries.

Experimental design

Thank you for your improvements. I have no further inquiries.

Validity of the findings

Thank you for your improvements. I have no further inquiries.

Additional comments

Thank you for your improvements. I have no further inquiries.